# Burning-derived vanillic acid in an Arctic ice core from Tunu, Northeastern Greenland

Mackenzie M. Grieman[1], Murat Aydin[1], Joseph R. McConnell[2], and Eric S. Saltzman[1]

[1]Department of Earth System Science, University of California, Irvine, Irvine, California, 92697-3100, USA
[2]Division of Hydrologic Sciences, Desert Research Institute, Reno, Nevada, USA

*Correspondence to:* Mackenzie M. Grieman (mgrieman@uci.edu)

**Abstract.** In this study, vanillic acid was measured in the Tunu ice core from northeastern Greenland in samples covering the past 1700 years. Vanillic acid is an aerosol-borne aromatic methoxy acid, produced by the combustion of lignin during biomass burning. Air mass trajectory analysis indicates that North American boreal forests are likely the major source region for biomass burning aerosols deposited to the ice core site. Vanillic acid levels in the Tunu ice core range from <0.005-0.08 ppb. Tunu vanillic acid exhibits centennial-scale variability in preindustrial ice, with elevated levels during the warm climates of the Roman Warm Period and Medieval Climate Anomaly, and lower levels during the cooler climates of the Late Antiquity Little Ice Age and the Little Ice Age. Analysis using a peak detection method revealed a positive correlation between vanillic acid in the Tunu ice core and both ammonium and black carbon in the north Greenland NEEM ice core from 600 to 1200 CE. The data provide multiproxy evidence of centennial-scale variability in North American high latitude fire during this time period.

## 1 Introduction

Proxy records of biomass burning are important for understanding the relationship between climate variability and fire on long time scales. A diverse range of paleoproxies has been used to reconstruct biomass burning, including fire scars on tree rings, sedimentary charcoal records, ice core records of gases and aerosol-borne chemicals (Blarquez et al., 2014; Legrand et al., 2016; Marlon et al., 2008, 2016; Power et al., 2008, 2013; Rubino et al., 2015). These records reflect a wide range of different aspects of fire location, frequency, distribution, and intensity. The proxies also integrate over a wide span of spatial and temporal scales. Despite these efforts, there is not yet a coherent global picture of historical variations in biomass burning and their relationship to climate over the past few millennia.

Northern Hemisphere burning trends on centennial time scales have been developed using composites of terrestrial sedimentary charcoal records (Marlon et al., 2008). These suggest that Northern Hemisphere burning declined during the Common Era, from about 1-1750 CE, then increased until 1870 CE. During the industrial period, burning rose from 1900-1950, declined from 1950-2000, and rose again after 2000 CE (Marlon et al., 2016). This composite record reflects conditions in Europe and North America, with few records from northern Asia. Greenland ice core ammonium, black carbon, and levoglucosan provide evidence for elevated North American burning prior to 500 CE and from 1000-1200 CE, although these records are not en-

tirely consistent with each other (Chýlek et al., 1995; Legrand et al., 1992, 2016; McConnell et al., 2007; Rubino et al., 2015; Zennaro et al., 2014).

A range of ice core proxies have been used to reconstruct biomass burning (Legrand et al., 2016; Rubino et al., 2015). Stable isotope ratios of ice core methane ($\delta^{13}CH_4$) have been used to infer global biomass burning, using end member isotopic com-positions of methane sources (Ferretti et al., 2005; Mischler et al., 2009; Sapart et al., 2012). Elevated ammonium concurrent with elevated levels of other chemicals has been used to reconstruct regional biomass burning. The difficulty of using ice core ammonium is that it is derived from several sources (Rubino et al., 2015). Ice core black carbon has been used as a tracer for preindustrial burning. Differences between these records could be due to variability in combustion conditions and transport (Rubino et al., 2015). The incomplete combustion of biomass produces organic aerosols. A large percentage of the biomass burning-derived organic aerosol is composed of levoglucosan, which is emitted by all plant matter containing cellulose (Si-moneit et al., 1999). However, levoglucosan has the potential for rapid degradation during atmospheric transport (Hennigan et al., 2010; Hoffmann et al., 2010; Slade and Knopf, 2013).

Aromatic acids in ice cores have also been used as ice core proxies for regional variability in biomass burning (Grieman et al., 2018). These compounds are produced from the pyrolysis of lignin and they retain the basic structure of the lignin backbone (Hedges and Mann, 1979; Hedges and Parker, 1976; Opsahl and Benner, 1995; Oros and Simoneit, 2001a, b; Simoneit, 2002; Vanholme et al., 2010; Akagi et al., 2011; Legrand et al., 2016). These compounds have been extensively studied in laboratory burning studies and atmospheric aerosols (Iinuma et al., 2007; Oros and Simoneit, 2001a; Oros et al., 2006; Otto et al., 2006; Rogge et al., 1998; Simoneit et al., 1999). These studies have shown that North American and European conifer and deciduous tree species produce vanillic acid (VA) (Simoneit, 2002; Oros and Simoneit, 2001a, b; Iinuma et al., 2007). VA is observed in atmospheric aerosols in the Arctic and Antarctic after long-distance transport (Zangrando et al., 2013, 2016). The lifetime of aromatic acids in the gas phase is on the order of a day due to oxidation by the hydroxyl radical. However, modeling studies suggest that in aerosols these compounds may be shielded from oxidation, resulting in atmospheric lifetimes of several days (Donahue et al., 2013). Laboratory and field studies have also shown decreased volatility of low molecular weight organic acids in aerosol form as a result of interactions with sea salt and other cations. Such studies have not yet been carried out on aromatic acids (Häkkinen et al., 2014; Laskin et al., 2012).

Until recently, relatively few measurements of these types of compounds have been made on ice cores. Recent studies include measurements of vanillic and p-hydroxybenzoic acid on cores from Greenland, Switzerland, the Kamchatka Peninsula, and Svalbard (Kawamura et al., 2012; McConnell et al., 2007; Müller-Tautges et al., 2016). The first continuous, millennial time scale ice core records of VA and p-HBA from Siberia and Svalbard have recently been published (Grieman et al., 2017, 2018).

In this study, VA was analyzed in an ice core from the Tunu region in Northeastern Greenland (Fig. 1). The ice core samples range in age over the past 1700 years (268-2013 CE) and this study presents the first continuous multi-centennial time scale ice core record of VA from Greenland. The Tunu record is compared to previous records of ammonium and black carbon from Greenland, to sedimentary charcoal records from likely source regions, and to patterns of historical climate change.

## 2 Methods

### 2.1 Ice core dating and sample collection

The Tunu ice core was drilled in 2013 to a depth of 212 m (Fig. 1; 78.04°N, 33.88°W, 2,000 m above sea level). The mean annual temperature at the Tunu-N weather station is -27.5°C (78.02°N, 33.99°W, 2,113 m above sea level; Steffen and Box, 2001). The ice core site has an average accumulation rate of 0.100 m water equivalent year$^{-1}$ (Sigl et al., 2015). The Tunu ice core age scale is based on the annual layer counting of non-sea salt sulfur and sodium signals (Fig. S1; Mernild et al., 2015). The Tunu time scale agrees to within $\pm 2$ years with the Northwest Greenland NEEM (NEEM-2011-S1) ice core based on comparison of the ages of major volcanic events (Mernild et al., 2015; Sigl et al., 2013, 2015).

Samples for this study were obtained from melting of a 33 x 33 mm cross section of the core (McConnell et al., 2001; Sigl et al., 2015; Mernild et al., 2015). Samples of the melt stream were collected continuously via fraction collector. Each sample has a time resolution of roughly 2.5 years.

### 2.2 Ice core sample analysis and data processing

Vanillic acid (VA) and para-hydroxybenzoic acid (p-HBA) were analyzed by anion exchange chromatographic separation and tandem mass spectrometric detection with electrospray ionization in negative ion mode (IC-ESI-MS/MS) (Grieman et al., 2017, 2018). The experimental method is described in detail in Grieman et al. (2017). The experimental system is a Dionex AS-AP autosampler, ICS-2100 integrated reagent-free ion chromatograph, and ThermoFinnigan TSQ Quantum triple quadrupole mass spectrometer. VA and p-HBA were detected at mass transitions of m/z 167→108 and m/z 137→93, respectively. Synthetic external standards, ranging in concentration from 0.1-2 ppb, were prepared using reagent grade VA and p-HBA in MilliQ water. These standards were analysed in sequence with ice core samples. The retention times of VA and p-HBA were 11.1 minutes and 11.8 minutes, respectively, with peak width half heights of 0.4 minutes. Detection limits for VA and p-HBA were 0.005 ppb and 0.034 ppb, defined as 3x the standard deviation of MilliQ water blanks (n = 58). 546 ice core samples from the Tunu ice core were analyzed in this study.

Tunu ice core VA ranges from below detection to 0.080 ppb, which is about 15x the detection limit (Fig. 2). The distribution of the measurements was skewed strongly towards lower concentrations, with VA levels below detection in 40% of the samples. The VA measurements below detection were replaced in the data set with a value of one half the detection limit, or 0.0025 ppb. The geometric mean VA level for the whole data set was $0.005^{+0.006}_{-0.003}$ ppb.

In analyzing the Tunu vanillic acid data set, a number of different methods were examined, including: 1) bin averaging of the log-transformed data (Grieman et al., 2017), 2) LOESS smoothing (Cleveland and Devlin, 1988), and 3) peak detection using singular spectrum analysis (Higuera et al., 2010; Fischer et al., 2015). The trends discussed here are robust in the sense that they are evident regardless of the data analysis method used. For example, similar patterns of centennial variability emerge from all of the methods (Fig. S2).

Although p-HBA is present in many samples, the levels are not reported due to interference at the m/z 137→93 mass transition eluting at nearly the same retention time (Fig. S3). This peak was present in most of the ice core samples, several of

the 58 MilliQ water blanks, and standards. The detection limit for p-HBA was determined using blanks that did not show this contamination. The presence of this peak in blanks and standards suggests that it is a contaminant that was introduced locally during sample handling. This peak was not observed in previous analyses of p-HBA in other Arctic ice cores (Grieman et al., 2017, 2018).

## 2.3 Air mass back trajectory analysis

Air mass back trajectory studies indicate that North America is the major source region for aerosols transported to the central Greenland ice sheet (Kahl et al., 1997). Here we examine transport from potential biomass burning source regions to the Tunu ice core site (78°N, 34°W) during the boreal burning season. 10-day air mass back trajectories were run from the Tunu ice core site at 12:00 AM and 12:00 PM local time (UTC-2 hours) beginning 100 m above ground level. Back trajectories were computed each day for spring (March 1-May 31), summer (June 8-August 31), and fall (September 1-November 30). The HYSPLIT model was used to compute the trajectories using NCEP/NCAR meteorological data for the ten-year period (2006-2015) (Kalnay et al., 1996; Draxler et al., 1999; Stein et al., 2015). Possible source regions were partitioned into ecofloristic zones in North America, Europe (defined as west of 42°E), and Siberia (defined as east of 42°E). Food and Agriculture Organization classifications were used to define ecofloristic zones in each of these regions (Fig. S6; http://cdiac.ornl.gov/epubs/ndp/global_carbon/carbon_documentation.html; Ruesch and Gibbs, 2008).

## 3 Results and Discussion

### 3.1 Tunu vanillic acid time series

Tunu vanillic acid (VA) levels do not exhibit a significant linear trend over the past 1700 years. They do exhibit pronounced variability on centennial time scales (Fig. 2). This variability is observable in the raw data as multi-decadal to century-long periods in which all of the measurements were above the detection limit. The variability is more clearly shown in the 40-year bin averages of the log-transformed data (Fig. 3). There is a maximum early in the record (280-400 CE) during the Roman Warm Period (RWP; 550 BCE-350 CE) followed by a period of generally lower levels from 500-1000 CE. A second maximum occurred during the Medieval Climate Anomaly (MCA; 1080-1240 CE) followed by generally declining levels through the Little Ice Age (LIA) and into the twentieth century. There is shorter-term (decadal or multi-decadal) variability throughout the record. A peak during the latter half of the twentieth century (1955-1985) stands out as the largest example of such variability in the record (Fig. 3, 4).

Burned area in the boreal forest of North America was high in the early 1900s, declined into the middle of the twentieth century, and began to increase again beginning in the 1960s (Mouillot and Field, 2005). The short-term period of elevated Tunu VA from 1955-1985 does not mirror the trend in this burned area record. A record of large boreal wildfires in Canada, with burned areas exceeding 200 ha, shows periods of large wildfires around 1960, the late 1970s to the early 1980s, around 1990, and the mid-1990s (Stocks et al., 2002). The similarity between the Tunu VA record and large boreal fire record from

Canada suggests that twentieth century VA record may be showing large fires. These large fires represent 3.1% of the number of Canadian fires from 1959-1997, but 97% of the area burned (Stocks et al., 2002).

We also examined variability in the Tunu VA record by analyzing the frequency of occurrence of highly elevated VA levels. This "peak detection" approach has been used to extract fire proxy signals from sediment charcoal and ice core ammonium records (Higuera et al., 2010; Legrand et al., 2016; Fischer et al., 2015). To apply this method, we used singular spectrum analysis (SSA) to decompose the Tunu VA record into 30 principle components (PCs). The PCs were converted to reconstructed components (RCs). RC-1 contains most of the low frequency content of the Tunu VA record. This component shows similar centennial-scale variability to the 40-year bin-averaged data, with elevated VA early in the record and around the MCA. It also emphasizes the decreasing trend from 1200-1900 CE and the increase during the 20th century (Fig. 5, top panel).

The higher frequency content of the VA record was reconstituted by summing RCs 2-30 (Fig. 5, middle panel). Individual peaks in the reconstituted record were detected as samples with VA levels exceeding a specified threshold. The threshold was defined as a percentile of the reconstituted data and peak frequency (peaks year$^{-1}$) computed in a moving 40-year window. Peak thresholds of the 65th, 70th, and 75th percentiles were used. The results are insensitive to the choice of threshold in this range (Fig. 5, lower panel). The peak detection algorithm shows strong centennial scale variability, in general agreement with the 40-year bin averaging. The amplitudes of the centennial scale features are relatively constant until the onset of the Little Ice Age, after which they appear to decline. At face value, the peak detection algorithm suggests that these centennial scale changes in Tunu VA involved not only variations in the abundance of VA, but also changes in the frequency of large fire events. In theory, comparing peak detection and bin-averaging signals could differentiate between biomass burning aerosols from distant fires that might be well-mixed throughout the Arctic atmosphere, as compared to large episodic inputs from major fire plumes in the immediate source regions impacting the site.

## 3.2 Relationship between Tunu vanillic acid levels and accumulation rate

Interpretation of ice core vanillic acid levels in terms of atmospheric aerosol concentrations or fluxes requires consideration of the processes by which aerosols are incorporated into the polar ice sheet. The concentrations of ions in polar ice cores likely reflect changes in the composition and abundance of the overlying aerosols, but may also be influenced by depositional and post-depositional processes that affect the air/snow transfer function (Grannas et al., 2007). We examined the relationship between VA and snow accumulation rate in order to assess the influence of variations in local conditions on Tunu ice core VA levels. VA flux was computed as the product of accumulation rate and VA concentration (F = $C_{ice}$ $A_{H_2O}$ $\rho_{ice}$). VA flux exhibits the same major features observed in the concentration record, indicating that the variations in VA likely reflect changes in atmospheric composition and are not caused primarily by changes in snow accumulation (Fig. 6). Excluding the 20th century, there is a positive correlation between VA flux and accumulation rate (r$^2$=0.25, p < 0.001) with slope of 0.011 $\pm$ 0.001 x 10$^{-9}$ kg m$^{-2}$ yr$^{-1}$ VA / kg m$^{-2}$ yr$^{-1}$ water flux (Fig. 7). This positive relationship is typical of ice core impurities, and may be interpreted in the context of a simple model of dry/wet deposition, such as:

$$F_{total} = F_{dry} + F_{wet}$$

$$= C_{air}V_d + C_{air}R_{scav}P_{H_2O},$$

where $F_{total}$ is the total VA flux, $F_{dry}$ is the VA dry deposition flux, $F_{wet}$ is the VA wet deposition flux, $C_{air}$ is the concentration of VA in the atmosphere, $V_d$ is a dry deposition velocity for the VA-containing aerosols, $R_{scav}$ is the wet deposition scavenging ratio, and $P_{H_2O}$ is the snow precipitation rate (Saltzman et al., 1997).

For a chemically stable, non-volatile species with variations in both atmospheric concentration and snow precipitation (that are not highly correlated), one would expect to observe: 1) a positive correlation between the depositional flux and water accumulation, and 2) a positive y-intercept reflecting the dry deposition component. A negative intercept could be interpreted as evidence of re-volatilization of VA from the snowpack, but the magnitude of this effect is evidently small.

During the 20th century, there is little change in accumulation rate associated with the very large increase in VA. This confirms that the 20th century increase is likely a real atmospheric feature and not an artifact related to local depositional conditions. If re-volatilization does occur after deposition, one would expect that loss of methoxy aromatic acids may increase with increased acidity. In this record, the opposite appears to be the case. Elevated levels of VA generally coincide with elevated levels of acidity, sulfate, and, to a lesser degree, nitrate during the 20th century (Fig. S4; S5). Increased acidity therefore does not appear to have induced significant loss of vanillic acid from the Tunu ice core.

### 3.3 Potential source regions for Tunu vanillic acid

The back-trajectories confirm that aerosols reaching the Tunu site predominantly originated in North America. North American trajectories comprised 32%, 17%, and 37% of the total for spring, summer, and fall, respectively (Table 1; Fig. S7). The most commonly transected North American ecofloristic zones were boreal tundra woodland and boreal coniferous forests. Trajectories from North American ecofloristic zones were more common in the fall and spring than in the summer. The percentages of trajectories from European and Siberian ecofloristic zones were all lower than 3% and 2%, respectively. The remainder of the trajectories originate over the oceans and did not transect North American, European, or Siberian ecofloristic zones. These results do not preclude wildfire emissions from Europe or Siberia reaching the Tunu ice core site but suggest that the frequency of such events is much lower than that for North American events. This result is consistent with Kahl et al. (1997), an earlier study of air mass transport to Summit, Greenland over a 40-year period (1946-1989).

### 3.4 Comparison to the NEEM Greenland ice core record of ammonium and black carbon

In order to develop confidence that ice core proxies of biomass burning are broadly representative of large-scale regional paleofire trends, it is important to demonstrate similar historical patterns from ice cores in the same geographic region and influenced by similar air mass trajectories. Levoglucosan and black carbon in the NEEM ice core from Northern Greenland are elevated from 200-600 CE and 100-700 CE, respectively (Fig. S8) (Zennaro et al., 2014). These periods overlap the period of elevated Tunu VA from 280-400 CE. They are still elevated 200-300 years after the Tunu VA record has declined. There is also an overlapping peak in the GISP2 ice core ammonium record from 320-330 CE (Chýlek et al., 1995). NEEM levoglucosan, black carbon, and ammonium are also elevated from 1000-1200 CE, 1000-1600 CE, and 1200-1500 CE respectively, at about

the same time as the peak in the Tunu VA record from 1080-1240 CE (Legrand et al., 2016; Zennaro et al., 2014). This peak is slightly earlier than a period of elevated ammonium, oxalate, and potassium in an ice core from the Eclipse ice field in western Canada from 1240-1410 CE (Yalcin et al., 2006). Periods of elevated burning in Greenland and Canadian ice core records after 1240 CE are not pronounced in the Tunu VA record. These periods include elevated NEEM levoglucosan from 1500-1700 CE,

20D and GISP2 ammonium from 1790-1810 CE and 1830-1910 CE, and periods of elevated burning in the Mt. Logan and Eclipse ice cores from western Canada in the 18th-20th centuries (Whitlow et al., 1994; Yalcin et al., 2006; Zennaro et al., 2014).

To further examine these relationships, we compare Tunu VA with ammonium and black carbon measurements on the NEEM ice core from North Greenland (Zennaro et al., 2014). Air mass trajectories indicate that the NEEM site should experience

similar transport to Tunu, with eastern Canada as the major fire source region followed by western Canada (Legrand et al., 2016). We first compare Tunu VA and NEEM black carbon using the first principle component of a singular spectrum analysis of each record (RC-1; Fig. 8). NEEM ammonium was not included in this analysis, as it is believed that long-term variations in ammonium levels reflect biogenic rather than pyrogenic emissions (Legrand et al., 2016; Fischer et al., 2015).

There are some similarities between Tunu VA and NEEM black carbon. Both records are low prior to the MCA and during

the LIA and both records are elevated during the RWP and MCA. There are also notable differences. For example, vanillic acid declines sharply into the Late Antique Little Ice Age (LALIA) around 450 CE, while black carbon and ammonium remain relatively high until 700 CE. Another difference is that black carbon reaches its LIA minima around 1750 and begins to rise thereafter. By contrast, VA continues its decline, reaching its minimum around 1900 CE. The divergence of these proxies during the industrial period is not surprising, as black carbon is emitted from fossil fuel combustion while VA is not. The divergence

between VA and black carbon during the industrial period was previously observed in a shallow central Greenland ice core (McConnell et al., 2007).

We next used the peak detection method described earlier in section 3.1 to examine the relationship of Tunu VA with NEEM ammonium and black carbon. A peak threshold of the 75th percentile was used for all records. Increasing the threshold from the 80th-95th percentiles reduces the number of peaks, but does not significantly alter their timing. For the data set as a whole,

all three of the fire proxies are positively correlated with one another (p<0.001). Visual inspection of the results shows that the relationships between Tunu VA, NEEM ammonium, and NEEM black carbon are complex, with periods of both positive and negative correlation. In order to better illustrate the relationships, we computed the correlation coefficient between the three pairs of proxies in a 200-year moving window (Fig. 8, lower panel). The results show a strong positive correlation for all of the records from 650-1200 CE but not subsequently. This abrupt change in relationship occurs around the same time as a major

change in North Atlantic climate discussed below (see section 3.6).

## 3.5   Comparison to charcoal records

We compared the Tunu ice core VA record to sedimentary charcoal records from North America. All charcoal records from Canada available in the Global Charcoal Database (106 records, 40°-80°N, 10°-160°W) were analyzed using the paleofire R package (http://gpwg.paleofire.org; Blarquez et al., 2014; Marlon et al., 2008). Regional Canadian charcoal records were

normalized using the Box-Cox, mini-max, and z-score transformations for 200-2000 CE, and composited using 40-year bin averages.

The Canada charcoal record exhibits neither a linear trend nor century-scale variability that is significant at the 95% confidence interval (Fig. S9). Canadian charcoal records show a low near 1600-1700 CE in the LIA and a high centered around 1000 CE early in the MCA. In the VA record, the high is around 1100-1200 CE and there is a gradual decline that reaches a minimum from 1800-1900 CE. While both records show a decline from the MCA to LIA, the timing does not match and there seems to be offset of 100-200 years between the maxima and minima in the two records. We also examined composite records for a number of smaller regions within Canada in an effort to identify variability similar to that in Tunu VA (Fig. S10). These regions are also not significant at the 95% confidence interval. Only the records from western Canada (40°-80°N, 110°-180°W) show any similarity to the Tunu VA record, with a slight increase around 1400. The lack of significant correlation between charcoal records and Tunu VA can in part be due to uncertainties in chronologies as charcoal record chronologies generally have much larger uncertainties than ice cores.

## 3.6 Relationship to climate

Over the past two millennia, Northern Hemispheric climate has been modulated by several centennial-scale climate anomalies: the Roman Warm Period or Roman Climate Optimum (550 BCE-350 CE; Wang et al., 2012), the Dark Ages Cold period (400-765 CE; Helama et al., 2017) or Late Antique Little Ice Age (536-660 CE; Büntgen et al., 2016), the Medieval Climate Anomaly (950-1250 CE; Mann et al., 2009), and the Little Ice Age (1400-1700 CE; Mann et al., 2009). The regional climate variability and dynamics associated with these anomalies are not well understood, particularly for the earlier Roman Warm Period and Late Antique Little Ice Age. During the Roman Warm Period, climate proxies from the North Sea, the Qinghai–Tibet Plateau, southwest Greenland, Spain, Iceland, and other Northern Hemisphere locations show increased climatic variability (Wang et al., 2012; Bianchi and McCave, 1999). Proxy records indicate that the Mediterranean experienced a wet and humid climate episode during this period (Wang et al., 2012). The Late Antique Little Ice Age is evident in tree ring records from the Russian Altai and European Alps. This period of cooling followed large volcanic eruptions and overlaps the Dark Ages Cold period that spanned the Northern Hemisphere (Büntgen et al., 2016). The contributing factors for this period are under debate, but may involve ice-rafting events, North Atlantic Oscillation, and/or El Nino-Southern Oscillation (Helama et al., 2017).

These climate anomalies appear to be reflected in the Tunu VA record, with elevated VA during the warm periods and lower levels during the colder periods (Fig. 9). Visual inspection of hemispheric mean temperature data suggests that elevated VA levels from 1080-1240 CE followed elevated Northern Hemisphere temperatures from about 970-1090 CE. Tunu $\delta^{18}$O, Pages 2k Arctic and North American temperature reconstructions show a similar relationship with VA levels (Fig. S11, S12) (McConnell, 2013; Pages2k, 2013). Elevated VA levels during the Roman Warm Period overlap elevated Arctic temperatures. Elevated VA levels from 1080-1250 CE follow elevated temperatures in the Arctic from about 930-1230 CE and North America from about 750-1150 CE. This relationship could be due to climate-driven changes in temperature or precipitation on burning extent, frequency, or location, as well as to changes in atmospheric transport patterns.

Skinner et al. (2006) examined relationships between Canadian forest fire season severity and prior winter global sea surface temperature variations from 1953-1999. They found three principle modes of influence: 1) the global near-linear trend in SST (i.e. Southern ocean warming and Atlantic and North Central Pacific cooling) positively correlated with fire severity across most of Canada, 2) the modulation of Atlantic SST associated with the Atlantic Multidecadal Oscillation (AMO) or Variability (AMV), with high fire severity across most of Canada associated with Atlantic cooling, and 3) variations in Pacific SST associated with PDO and ENSO, with high fire severity in western Canada and low severity over southern and eastern Canada associated with warm Pacific SST. Based on these results, one might speculate that externally forced climate variability due to orbital variations or volcanic eruptions might modulate Canada fire emissions via the global SST mode, while internal variability in the AMV and PDO modes might dominate on multi-decadal time scales (60-80 years). We examined several proxy reconstructions for these modes in order to test these relationships.

The PAGES Oceans 2K reconstruction shows a monotonic cooling trend for the global ocean over the past two millennia, with the exception of warming during the past century (McGregor et al., 2015). A notable exception to the trend is a period of warming in the Atlantic from 1000-1300 CE during the MCA (Sicre et al., 2008, 2011, Fig. 9), while Pacific SST maintained its slow cooling trend. Proxy records indicate that both the AMV and the North Atlantic Oscillation were in the positive phase during this period, consistent with a warm Atlantic (Meeker and Mayewski, 2002; Trouet et al., 2009; Olsen et al., 2012; Wang et al., 2017). Surprisingly, the observation of elevated Tunu VA during the warm MCA suggests a relationship between North American fire and SST that is opposite in sign to that inferred from analysis of the latter half of the 20th century by Skinner et al. (2006). That could indicate that different modes of variability influence North American fire on centennial and decadal time scales.

## 4  Conclusions

The Tunu ice core vanillic acid (VA) record is a new centennial time-scale record of burning emissions predominantly originating from the mid-high latitude forests of North America. At this stage, ice core VA should be regarded as a qualitative tracer because it is not known to what extent the signals reflect paleofire emissions, paleofire frequency, or changes in air mass transport and deposition. Further work comparing VA in shallow ice cores to satellite measurements and modeling of fires during recent decades would improve our understanding of the origins of the VA signals in Greenland ice.

The correlation between Tunu VA and ammonium and black carbon in the NEEM ice core from 600 to 1200 CE is evidence of centennial-scale variability in North American high latitude fire during this time period. Further measurements on multiple ice cores will be needed to validate this conclusion. A clear link between the VA variability in Greenland ice and North American sedimentary charcoal is not evident. The relationship between Tunu VA and multi-century climate anomalies suggests that on long time scales, North American burning may be positively correlated with Atlantic sea surface temperatures–a relationship that is not typical of the 20th century. The forcing and internal dynamics associated with late Holocene centennial scale climate variability remain a subject of debate and active research. Understanding those dynamics will be essential in order to unravel the fire-climate relationship.

# 5  Data availability

The data reported in this manuscript have been submitted to the NSF Artic Data Center (http://arcticdata.io/) (Grieman and Saltzman, 2018). NEEM ammonium and black carbon data and Tunu sulfate, acidity, and nitrate data were obtained from the Arctic data Center (McConnell, 2013, 2016). Northern Hemisphere temperature proxy and instrumental data were obtained

from NOAA

(www.ncdc.noaa.gov/paleo/study/6252; Mann et al., 2008). AMV data were obtained from NOAA (www.ncdc.noaa.gov/paleo-search/study/22031; Wang et al., 2017). The SST reconstruction was accessed through NOAA (ftp.ncdc.noaa.gov/pub/data/paleo/contributions_by_author/sicre2011/sicre2011.txt; Sicre et al., 2011). NAO data were accessed through NOAA (ftp.ncdc.noaa.gov/pub/data/paleo/treering/reconstructions/nao-trouet2009.txt; Trouet et al., 2009). Re-

gional temperature reconstructions were accessed through NOAA (www.ncdc.noaa.gov/paleo-search/study/14188 Pages2k, 2013).

*Author contributions.*  Mackenzie Grieman and Eric Saltzman drafted the manuscript. The analytical technique was developed by Mackenzie Grieman, Murat Aydin, and Eric Saltzman. Joe McConnell drilled and processed the ice core. Mackenzie Grieman and Joe McConnell collected the discrete melt water samples. Mackenzie Grieman analysed the melt water samples and processed the data set. Murat Aydin and

Joe McConnell provided comments on and edited the manuscript.

*Competing interests.*  The authors declare that they have no conflicts of interest.

*Acknowledgements.*  NCEP/NCAR reanalysis data were accessed from: ftp://arlftp.arlhq.noaa.gov/pub/archives/reanalysis. We would like to acknowledge M. Sigl, O. Maselli, N. Chellman, and R. Rhodes for collecting the Tunu core and for their help in obtaining the ice core melt water samples, C. McCormick for assistance with instrument maintenance, and T. Sutterley for coding assistance. We would also like to

acknowledge research support by a generous donation from the Jenkins Family to the Department of Earth System Science, University of California, Irvine. The Tunu core was collected and the primary analyses completed with support from National Science Foundation grant PLR-1204176 to JRM. Funding was also provided by the National Science Foundation (grant ANT-0839122; PLR-1142517) and by the NSF Independent Research/Development program.

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

**Table 1.** Percentages of air mass back trajectories crossing-over or reaching various geographic regions and ecofloristic zones beginning at the Tunu drilling site (% rounded to nearest integer). Ecofloristic zones are defined using Food and Agriculture Organization classifications (Fig. S2; http://cdiac.ornl.gov/epubs/ndp/global_carbon/carbon_documentation.html; Ruesch and Gibbs, 2008).

| Geographic region | Ecofloristic zone | Season | | |
| | | Spring | Summer | Fall |
|---|---|---|---|---|
| North America | boreal tundra woodland | 10 | 3 | 14 |
| | boreal coniferous forests | 6 | 2 | 8 |
| | other | 16 | 12 | 15 |
| | total | 32 | 17 | 37 |
| Europe | total | 4 | 2 | 4 |
| Siberia | total | 3 | 1 | 4 |

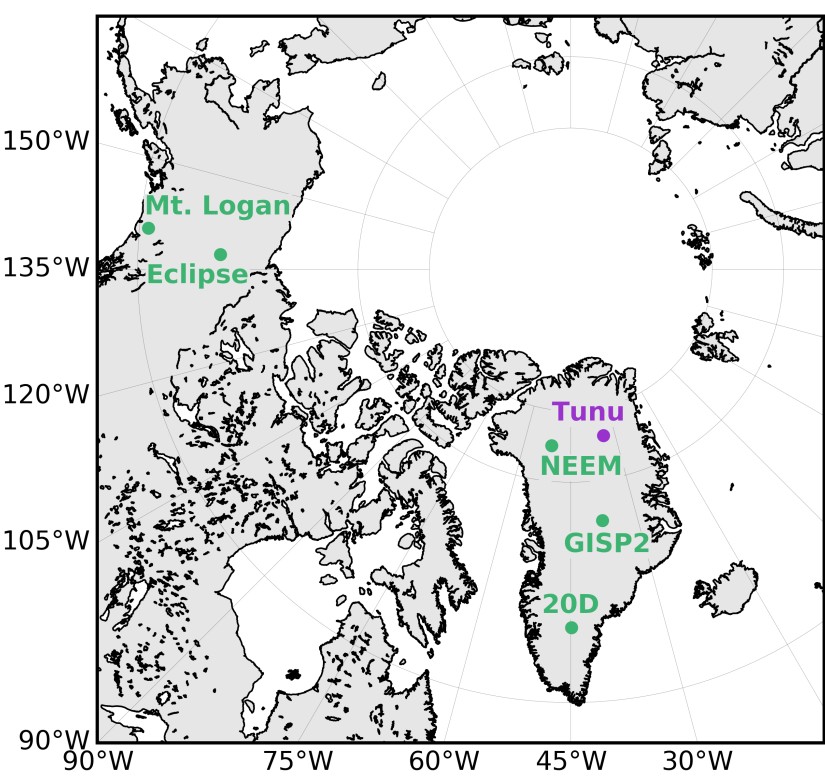

**Figure 1.** Tunu (78.04°N, 33.88°W, this study), NEEM (77.49°N, 51.2°W), GISP2 (72.2°N, 37.8°W), 20D (65.01°N, 44.87°W), Eclipse Icefield (60.51°N, 139.47°W), and Mt. Logan (60.58°N, 140.58°W) ice core drilling sites (Sigl et al., 2015; Whitlow et al., 1994; Yalcin et al., 2006; Zennaro et al., 2014).

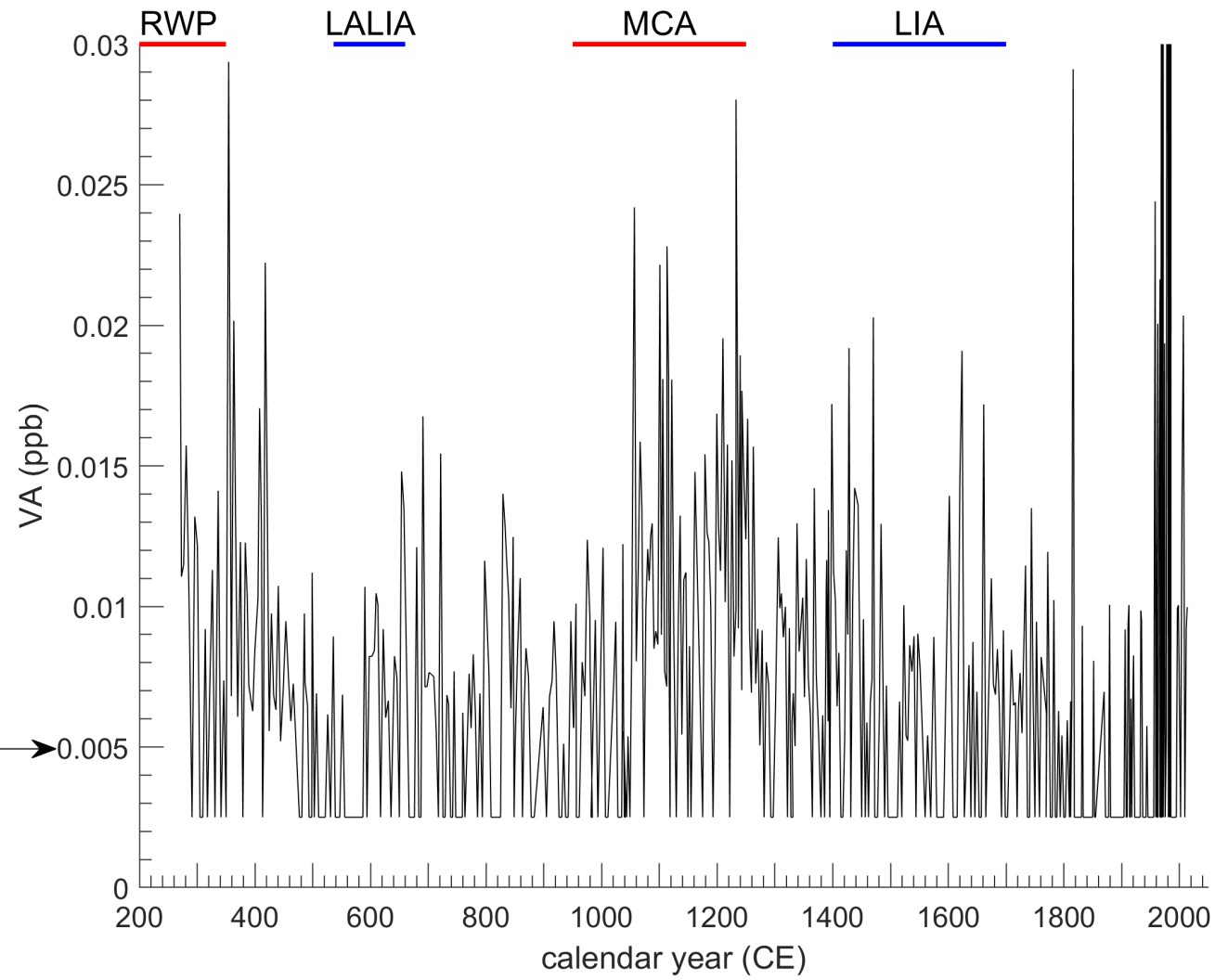

**Figure 2.** Tunu ice core vanillic acid (VA) record. The horizontal lines show the Roman Warm Period (RWP), Late Antique Little Ice Age (LALIA), Medieval Climate Anomaly (MCA), and the Little Ice Age (LIA) (Büntgen et al., 2016; Mann et al., 2009; Wang et al., 2012). The arrow indicates the limit of detection. All data below the limit of detection are treated as half of the limit of detection. Six measurements above 0.03 ppb not shown in this figure are shown in Fig. 4.

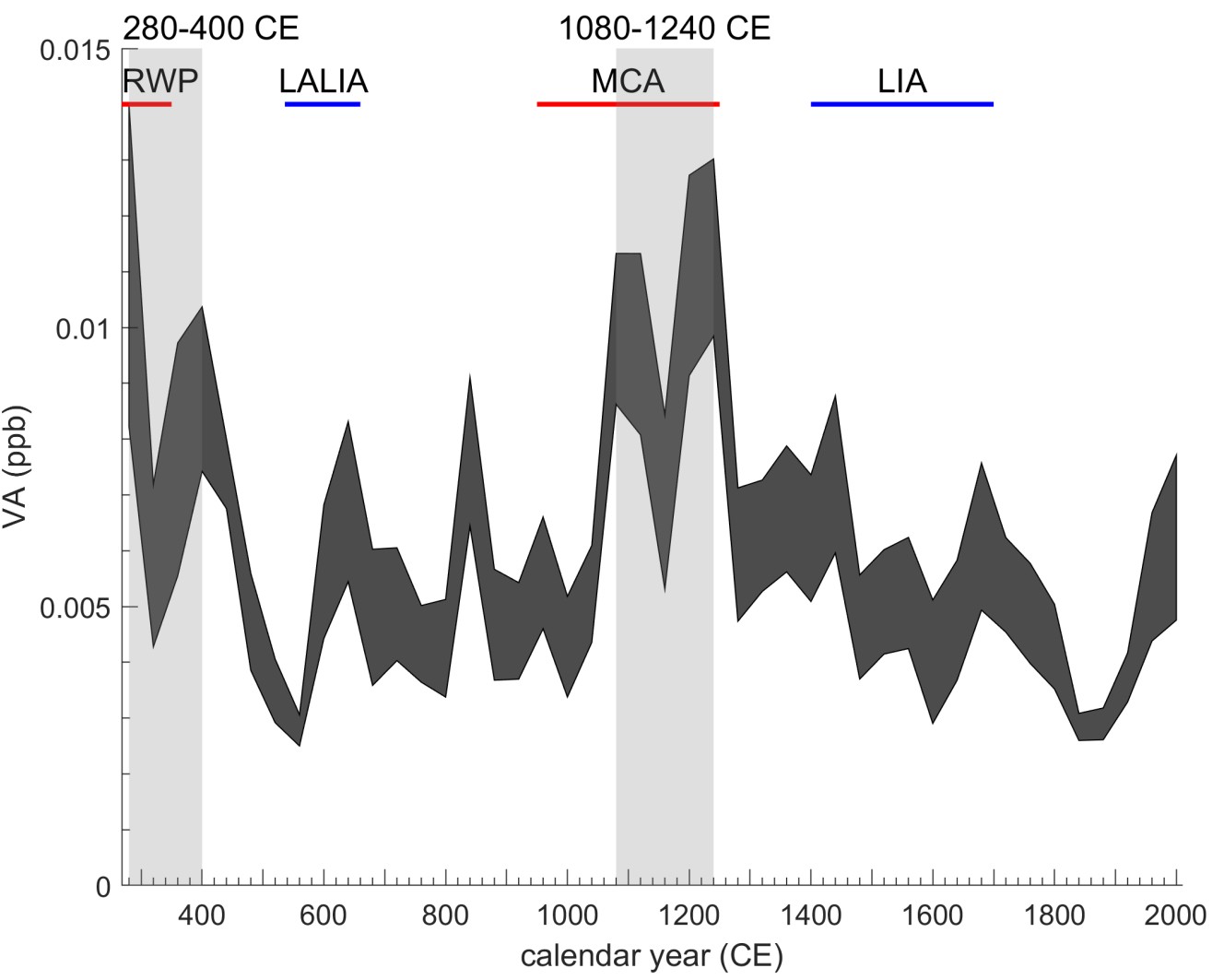

**Figure 3.** Tunu ice core vanillic acid (VA) record. The gray filled lines are ± 1 standard errors of the 40-year bin averages of the data. The horizontal lines show the Roman Warm Period (RWP), Late Antique Little Ice Age (LALIA), Medieval Climate Anomaly (MCA), and the Little Ice Age (LIA) (Büntgen et al., 2016; Mann et al., 2009; Wang et al., 2012). Light gray shaded areas are elevated periods in the record.

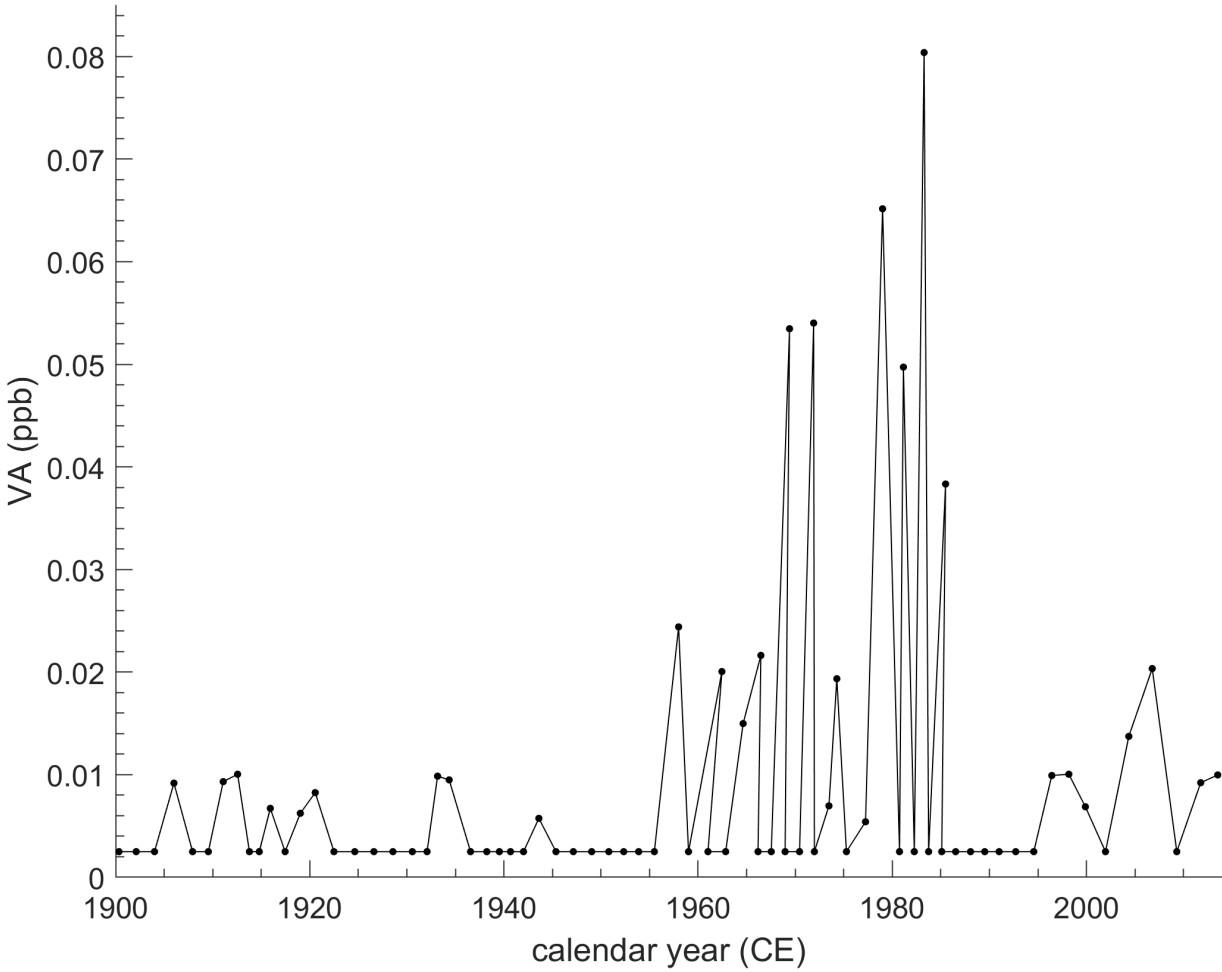

**Figure 4.** 20th century Tunu ice core vanillic acid (VA) record.

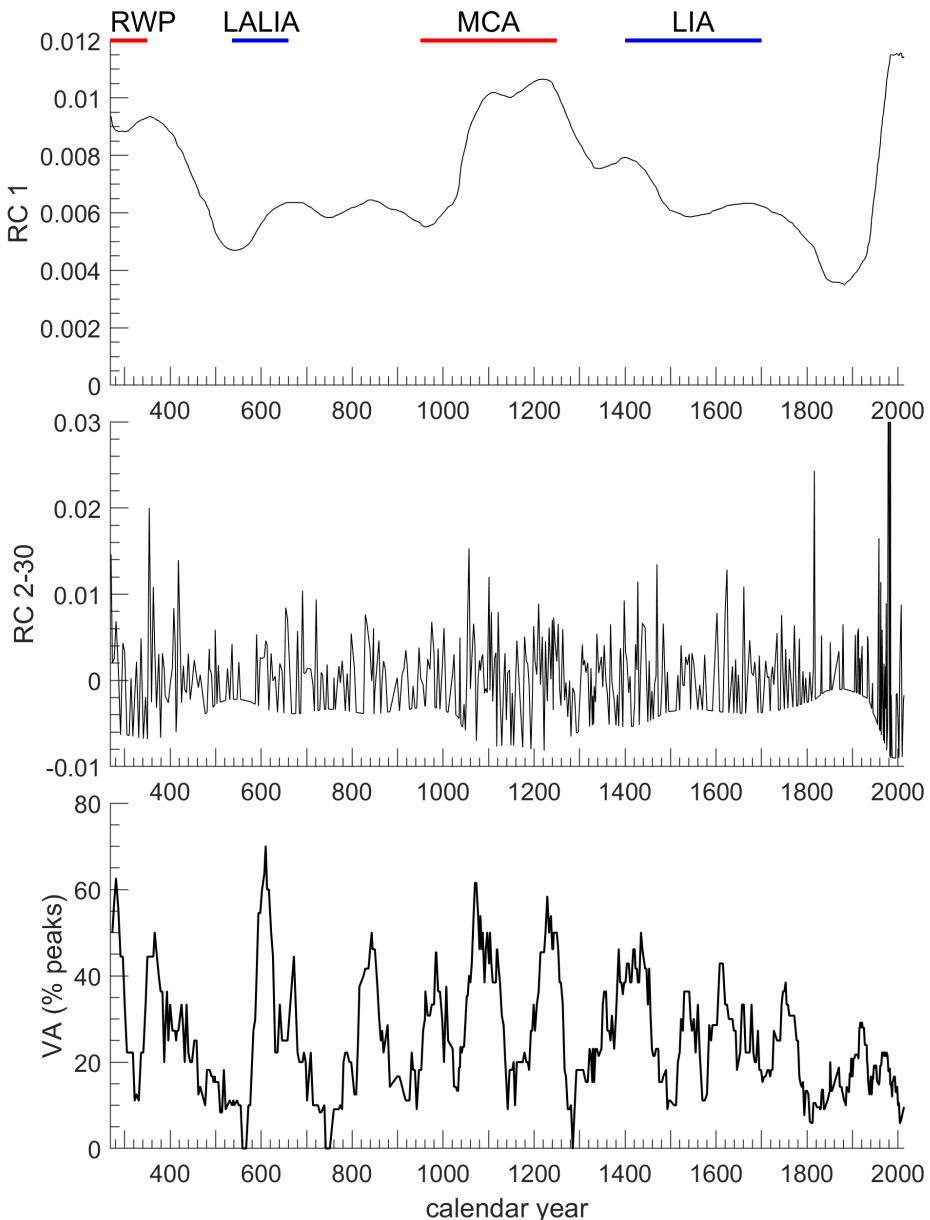

**Figure 5.** Tunu ice core vanillic acid analyzed by singular spectrum analysis (SSA) and peak detection showing centennial scale variability. 1) Low frequency variability in Tunu vanillic acid reconstructed using component 1 of the SSA, 2) higher frequency variability in Tunu vanillic acid reconstructed using SSA components 2-30, excluding component 1, 3) Peaks detected in the higher frequency Tunu vanillic reconstruction (components 2-30), shown as the fraction of ice core samples exceeding a peak threshold (75th percentile) in a 40-year running window. The horizontal lines at the top show the Roman Warm Period (RWP), Late Antique Little Ice Age (LALIA), Medieval Climate Anomaly (MCA), and the Little Ice Age (LIA) (Büntgen et al., 2016; Mann et al., 2009; Wang et al., 2012).

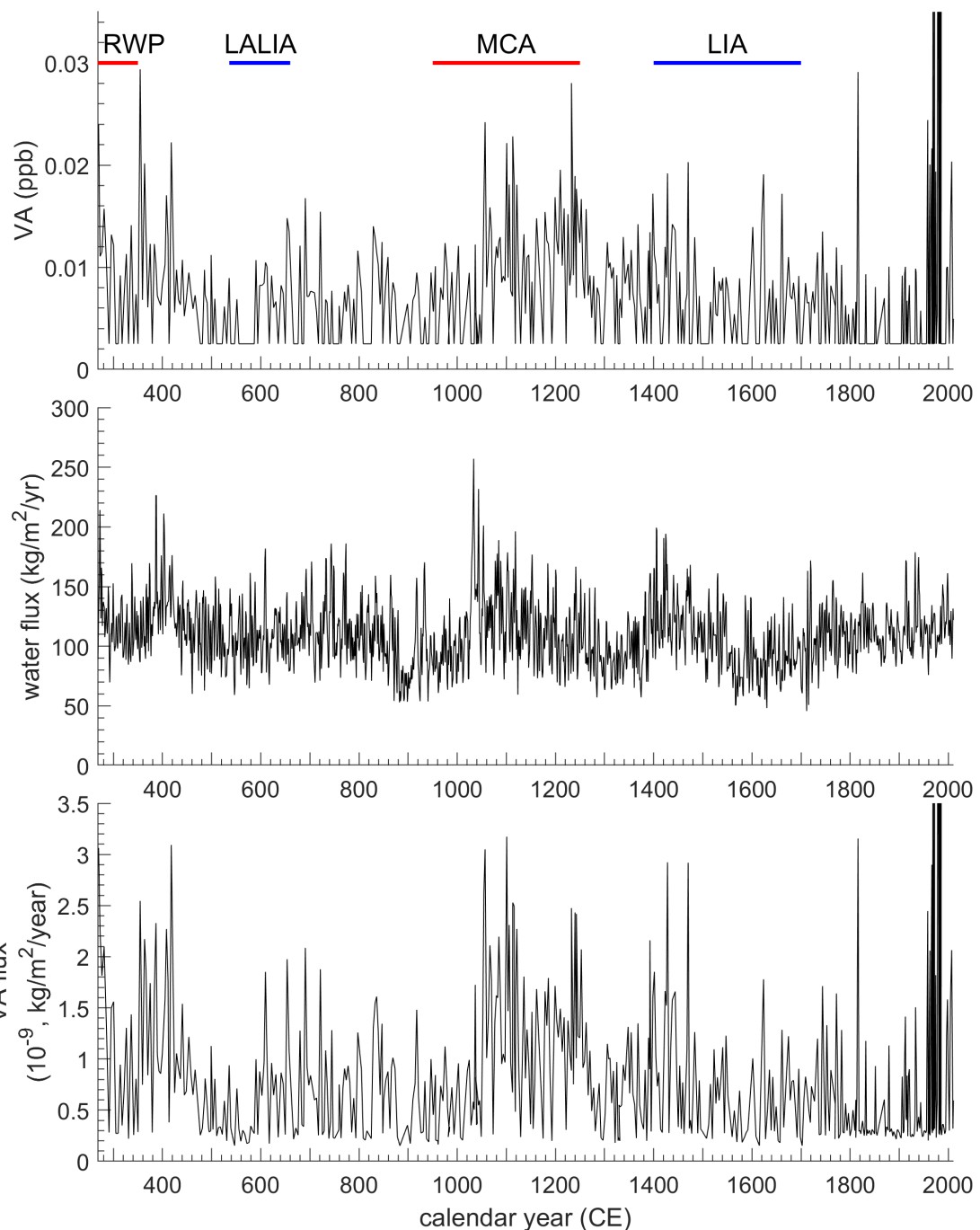

**Figure 6.** Tunu ice core vanillic acid concentration (top), water accumulation flux (middle), and vanillic acid depositional flux (bottom) computed as the product of vanillic acid concentration and water accumulation flux. The horizontal lines at the top show the Roman Warm Period (RWP), Late Antique Little Ice Age (LALIA), Medieval Climate Anomaly (MCA), and the Little Ice Age (LIA) (Büntgen et al., 2016; Mann et al., 2009; Wang et al., 2012).

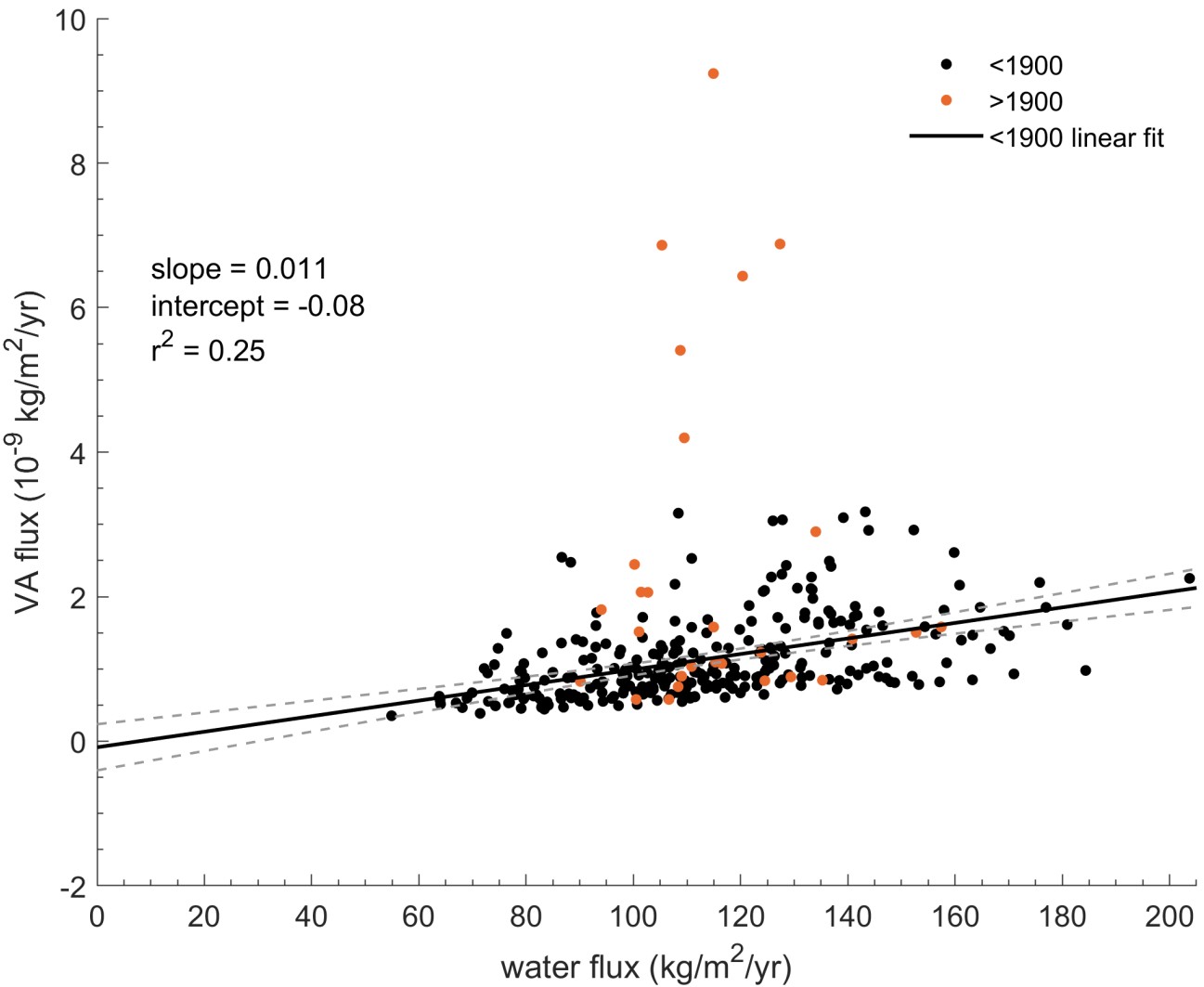

**Figure 7.** Relationship between Tunu vanillic acid depositional flux and water accumulation flux with linear regression to samples with ages older than 1900 CE (black points and line). Data younger than 1900 are shown in orange. Gray dashed lines show the uncertainty in the slope and intercept. Data below the limit of detection are not included.

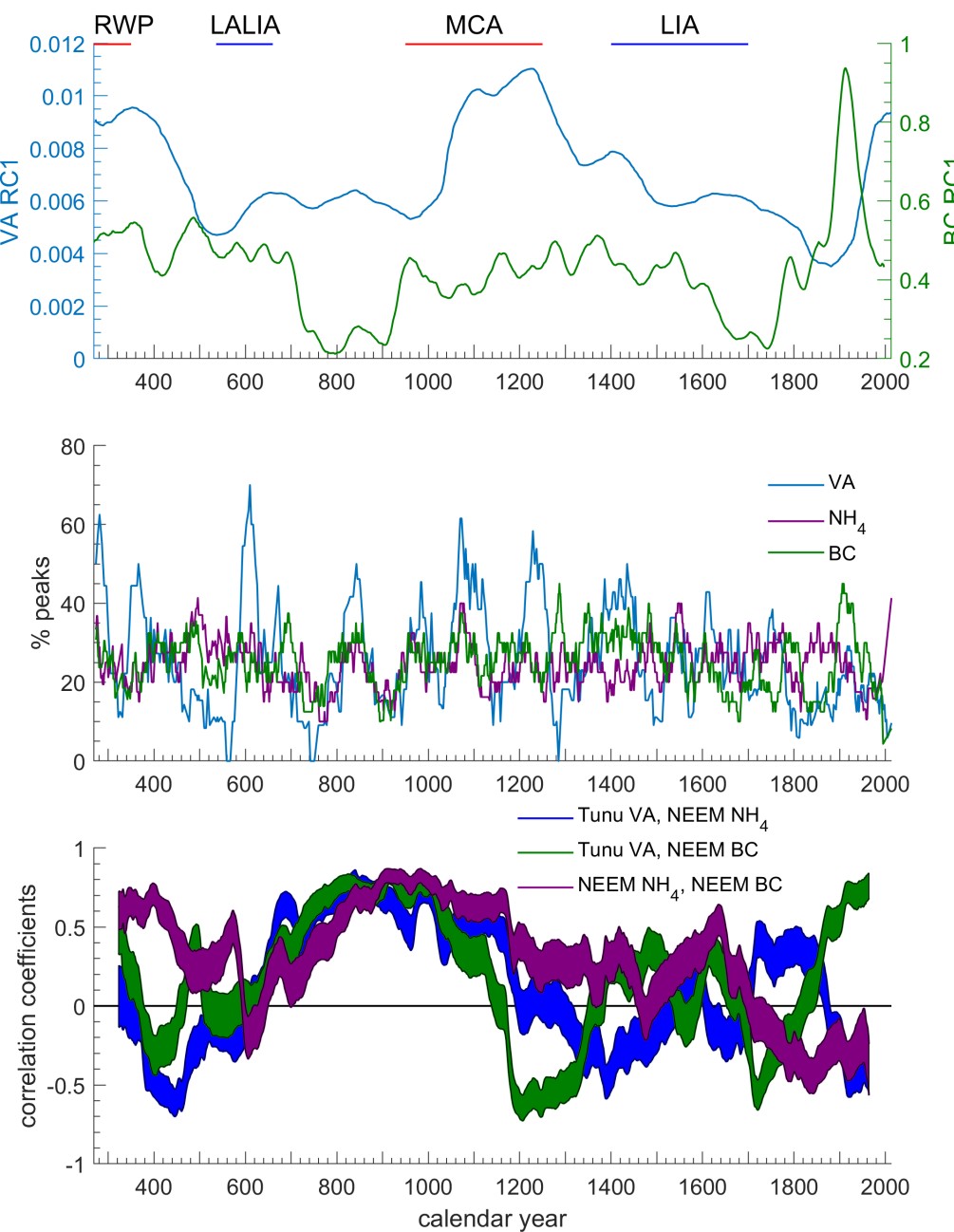

**Figure 8.** Relationships between biomass burning proxies in two north Greenland ice cores for the past 1700 years: Tunu ice core vanillic acid (VA), NEEM ammonium ($NH_4$) and NEEM black carbon (BC). 1) First component from the singular spectrum analysis of the three ice core signals (PC-1), 2) Frequency of peaks in the ice core signals reconstructed using singular spectrum components 2-30 and peak threshold of 75th percentile, smoothed with a 40-year running window, 3) 95% confidence intervals of correlation coefficients for the ice core peak frequencies using a 200-year running window (p<0.001). The horizontal lines at the top show the Roman Warm Period (RWP), Late Antique Little Ice Age (LALIA), Medieval Climate Anomaly (MCA), and the Little Ice Age (LIA) (Büntgen et al., 2016; Mann et al., 2009; Wang et al., 2012).

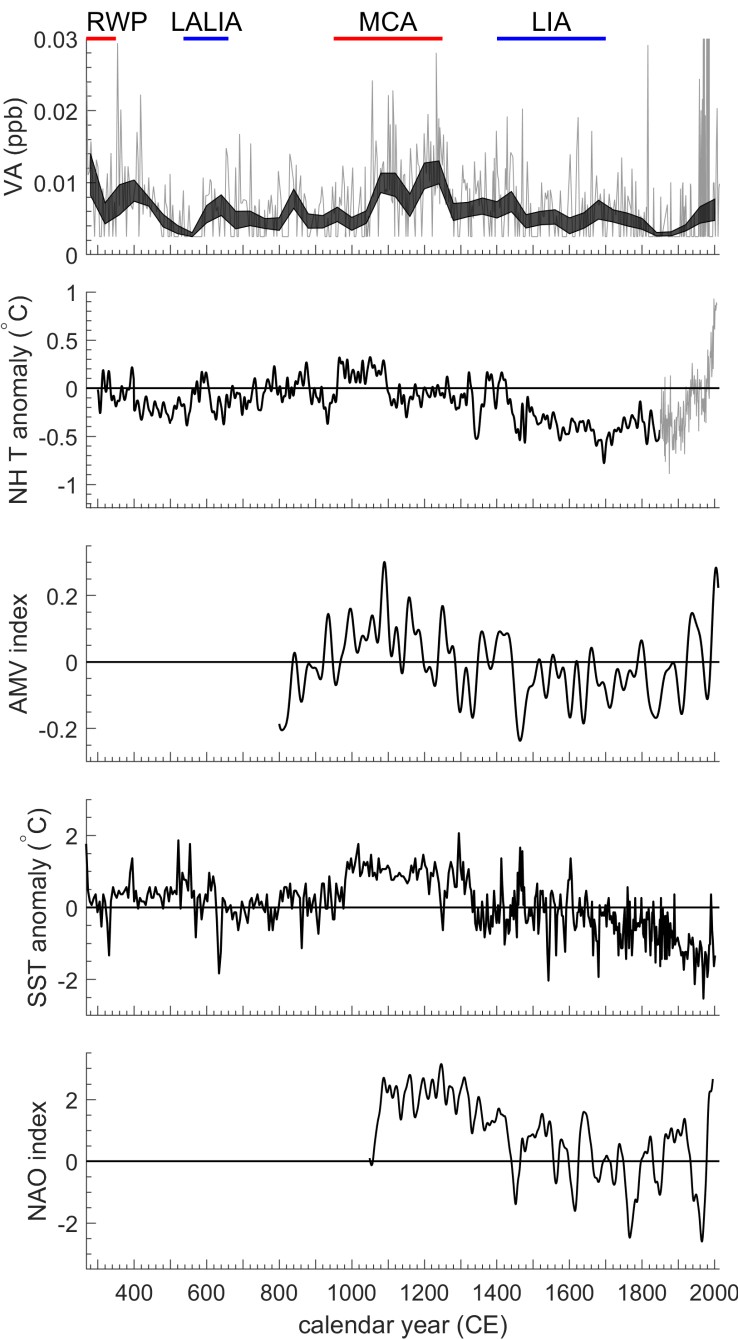

**Figure 9.** Comparison between Tunu vanillic acid (VA) and climate records. From top: Tunu VA, the gray filled lines are ± 1 standards errors of the 40-year bin averages of the data; Northern hemisphere land temperature anomaly composite before 1850 (EIV method; black line) and after 1850 (CRU instrumental record; gray line) (Mann et al., 2008); 30-year low-pass filtered Atlantic multidecadal variability (AMV) (Wang et al., 2017); North Atlantic alkenone sea surface temperature (SST) anomaly based on MD99-2275 sediment (Sicre et al., 2011); and North Atlantic Oscillation (NAO) index reconstruction (Trouet et al., 2009).