# Peer review of "Burning-derived vanillic acid in an Arctic ice core from Tunu, Northeastern Greenland"

_Climate of the Past, 2018_

## Referee Comment (RC1) · Anonymous Referee #1 · 19 May 2018

The authors present a valuable record of vanillic acid (VA) from the Tunu ice core in northeastern Greenland. This record is part of a growing suite of high latitude biomass burning records in recent years that have been analyzed by the authors as well as other research groups. These data are an important contribution to the paleoclimate, ice core and fire science communities. This work fits squarely within the goals of "Climate of the Past".

Unfortunately, the paper seems as if the authors wrote the manuscript in a rush. The authors stretch to find correlations where these relationships are not clear, and only briefly discuss any possible climatic influences (Section 3.6). The authors do not compare the Tunu VA record to any other proxy information derived from the Tunu core, other than accumulation, which is surprising as the ice core data would help to bet-

ter link the fire activity recorded at Tunu to climate variables influencing the site. I understand that we often have deadlines by which to submit papers in order to meet outside goals, which may be the reason for a possible rush. However, this paper needs substantial revision as the current form contains multiple interpretations that are not supported by the data.

Section 2.2: The authors mention that they do not report p-HBA in any of the samples due to interference at the m/z 137-> 93 mass transition, and ascribe this interference to a possible contamination during sample handling and/or storage. However, this interference is also present in the blanks. Are these blanks the 58 MilliQ water blanks that were previously mentioned? Does the interference also occur in the procedural blanks? (It is unclear if you have both lab and procedural blanks. Please specify exactly how you created your blanks). This peak also occurs in your standards, suggesting that the source of contamination is local, yet you are able to quantify detection limits for p-HBA. Please explain.

Section 2.2: I can understand the want to just site a previous method study to then move on more quickly to the results and discussion. However, more information from the Grieman et al., 2015 method paper is necessary within Section 2.2 of this paper. (Please also change your citations from Grieman et al., 2017, 2018, in review, to Grieman et al., 2015 and 2017 to correctly cite to work to which you refer). For example, what QA/QC measures do you use? Do you apply an internal standard? Readers cannot analyze your interpretations of the data without having a clear indication of the quality of the data.

Section 3.3: If 17% of the back-trajectories are from North America in the summer, and if 3% are from European ecofloristic zones, and 2% from Siberian ecofloristic zones, where is the source of the rest of the summer back-trajectories? I was hoping the Figure S7 would help determine where the other 78% of the summer back-trajectories originate, but this is not the case. It would be good to demonstrate in another figure in the Supplementary Information the paths of the other summer back-trajectories, as

well as the back-trajectories of the 3% from Europe, and 2% from Siberia.

Section 3.4 and Figure 8: You mention that "the results show a strong positive correlation for all the records from 650-1200 CE and a more variable and weaker relationship thereafter". The approach of computing the "correlation coefficient between the three pairs of proxies in a 200-year moving window (Fig. 8, lower panel)" is an excellent way to demonstrate the relationships between these three proxies. However, while the relationships between these proxies after 1200 CE certainly are variable, and do not demonstrate any patterns either between themselves or with each other. The only relationship that does seem to exist – and this is really quite stretching the interpretation – is that the correlation coefficients for Tunu VA, NEEM NH4 versus Tunu VA, NEEM BC are out of phase with one another after 1300 CE. It is better to state that after 1300 CE no major relationships exist between these data. Panel 2 is difficult to read. Either separating each proxy or elongating the y-axis will help the reader.

Section 3.5 and Figure S9: You mention "the qualitative similarities between the Tunu VA and western Canadian charcoal record suggest that this may be a source region of the Greenland burning signals, but the correlation is not significant at the 95% confidence interval." However, in examining Figure S9 closely, the only correlation occurs at 1400 CE, while all other peaks and low points are offset from one another. These records do not correlate, and ascribing western Canada as a source region based on this data is overreaching what the data actually demonstrate.

Section 3.6 and Figure 9: "The data suggest a positive correlation between North American fire and hemispheric mean temperature." This statement assumes that Tunu VA represents North American fire rather than boreal fire and/or regional fire. The comparison between Tunu VA and temperature for the entire northern hemisphere differs from your plots showing that the back-trajectories are primarily for high latitude regions. Using a temperature record that reflects your source region is a better comparison. (Using a representative temperature record is especially important as the MCA and LIA vary regionally, and therefore influence your time periods of interest).

Did you calculate this correlation? Or do the data only "suggest" this correlation? The peak in Tunu VA and the peak in the NHT anomaly are offset by at least a century. The decreased temperature and lower concentrations of Tunu VA do occur during similar time periods.

Conclusions and abstract: The correlation between Tunu VA, and Neem ammonium and black carbon in the NEEM ice core only exists between 600 to 1200 CE, and so following statement is misleading: "The correlation between Tunu VA, and NEEM ammonium and black carbon in the NEEM ice core is encouraging evidence that a consistent pattern of centennial-scale variability in North American high latitude fire is recorded in Greenland ice, but further measurements on multiple ice cores will be needed to validate this conclusion". A similar sentence is also present in the abstract. This correlation is only true for one part of the record, but absolutely does not apply to the rest of the record.

Conclusions: The statement "A clear link between the VA variability in Greenland ice and North American sedimentary charcoal is not evident, although a tentative connection to the Quebec region was noted". This statement contradicts your text in Section 3.5 in which you highlight possible similarities (with which the data do not actually demonstrate, as mentioned in the above comment addressing Section 3.5) between the Tunu NA and western Canadian charcoal syntheses. It is essential to omit the conclusion regarding Quebec.

Figure 5: The information given by the fraction of ice core samples exceeding the 70th percentile actually detracts from highlighting the difference between the fraction of ice core samples exceeding the set peak threshold in the 65th and 75th percentiles. However, as you use the 70th percentile in section 3.4 and as the basis for the second and third panels of Figure 8, it may be better to only show the 70th percentile. These colors (as in Figure 8) are unfortunately the most difficult to differentiate for people who are color blind, resulting in a figure that is unreadable for many people. Please choose other colors.

Miscellaneous: (Page numbers apply to the PDF version available to reviewers)

Page 3 Lines 32 and 33 (and continuing throughout the paper): PC's and RC's should be changed to PCs and RCs. As these components are not possessive, they do not require an apostrophe.

Page 4, Line 11: Place "of" between "frequency large".

Page 4, Line 12: Change "fiers" to "fires".

Page 4, Line 15: Omit the comma after "fluxes".

Page 6, Line 25: "The Canada charcoal record exhibits no linear trend and century scale variability that is not significant at the 95% confidence interval". This is double negative. Do you mean "The Canada charcoal record exhibits neither a linear trend nor century-scale variability that is significant at the 95% confidence interval"?

---

## Referee Comment (RC2) · Anonymous Referee #2 · 14 Jun 2018

This study presents new interesting measurements of vanillic acid as a proxy for biomass burning from an ice core record in northeastern Greenland spanning the past 1700 years. The authors examine the variability in the measurements in light of previously identified changes in climate and fire activity. Questions are posed about the source area of the VA, and the controls on its variability; these are addressed through air mass back trajectory analysis and through comparisons with data from other records and proxies.

The new study represents an important contribution to the literature and the paper is generally clear. Various sections, however, need more careful attention, especially because the data and analyses tend to raise more questions than they answer (although this is often the case, and especially when developing new proxies for environmental

changes). However, in this case there should be some discussion of what in particular might be most informative/useful going forward in terms of new proxy comparisons, or modeling, etc. Nor is there discussion of the more recent variations and how they compare with historical data, which is a large omission that needs to be addressed, especially given the 2.5-year temporal resolution of the ice core record noted in Section 2.1. Furthermore, how might we better understand the source areas or relationships between VA and other fire proxies from ice (e.g. what benefits does VA provide in comparison with BC, ammonium, levoglucosan, etc.)? Where are the greatest sources of uncertainty? Clarifying the limitations of the existing study and adding some thoughts about 'next steps' would be helpful for those outside the ice core community in particular.

In general, the conclusions are cautious and generally aligned with the data presented. VA holds some promise insofar as it appears consistent with other proxies in showing high fire activity during the RWP and MCA in particular, and low during the LILIA and LIA. With more careful attention to detail, some reorganizing, and some figure clean-up this will make a useful contribution to our understanding of how fire has changed on long time scales and what can be inferred from different proxies.

A few sections in particular need attention.

Section 3.3 – This section is describing methods but is located in the Results section – it seems more appropriate to methods. In addition, the 17% of the summer back trajectories reaching North America is quite low – where do all the trajectories that are not accounted for come from?

Section 3.6 also needs work. The methods referenced are entirely absent from the "Methods" section again and need to be moved and expanded there to understand what exactly was done (Pg. 7 line 19 is insufficient). Moreover, the influence of different modes of climate variability on the VA record are nearly impossible to disentangle given that the decadal-scale variability in the VA record raises more questions than it

addresses – it seems unclear still what exactly is producing the short-term variability in VA (in terms of fire), let alone what would cause the variations in fire themselves IF that is what the VA variations are reflecting. Furthermore, the features that are most robust in the VA record – high values during the RWP and MCA, and low values during the LALIA and LIA – are still not well understood. The RWP itself is not a well-known or widespread climate feature, so more information and background about that could be provided instead of discussion of possible climate modes that might control fire. I would rather see more investigation into understanding the potential relationships between the VA record and other fire proxies than a detailed investigation into the interannual climate controls on a very uncertain biomass burning reconstruction.

Detailed comments:

Pg 2. VA is described as resulting from the combustion of lignin. McConnell et al. (2007) indicated that conifers in particular are expected to be the primary source of VA – it seems worth noting this again, and also providing more information that might be useful based on the extensive previous research on VA that is cited. E.g. what is the expected atmospheric residence time of VA?

Pg 4 Line 6: Were higher thresholds tried? Would thresholds in the 80s or 90s produce very different results? Why did you choose this range?

Pg 4 Line 8: remove "Clearly" and explain what "these" refers to.

Pg 4 Line 12: should say "fires"

Pg 4 line 20: define the terms in the equation and make your words consistent with the equation.

Pg 4 Lines 26-80 – the equation and its description are not rendering properly (esp. check the capitalization and subscripting; vd, rscav).

Fig. 4 – Please discuss this figure and how it relates to the known fire history of NA. Fire history data are available from tree-ring (e.g. Giardin's papers, the Canadian

Large Fire Database) and historical studies and the data raise important concerns that warrant more discussion. Specifically, the high VA values in the 1960's and 70's are puzzling given that this pattern is directly opposite what historical records show (e.g. Mouillot and Field, 2005; Mouillot et al. 2006), where burning was very low in the middle part of the century. Many more recent NA fire records generally show that burning has increased most rapidly from low levels since the 1980's (NIFC, Littell et al. 2009). Is it possible that VA could at times have been influenced by some intensive industrial or lumber/logging-related processes that might have been occurring during this time in NA? In any case, in light of these data it is good that the authors state that VA is best considered a qualitative proxy at this stage and requires further exploration – this acknowledgement is appreciated.

The first reviewer raised many good concerns about the correlations presented, I trust those will also be addressed.

---

## Author Comment (AC1) · 31 Jul 2018

The referee made several comments that are much appreciated. The manuscript has been updated as follows to take them into account.

1. "only briefly discuss any possible climatic influences (Section 3.6)."

The following has been added to paragraph 1 in Section 3.6: "Climate proxies from the North Sea, the Qinghai–Tibet Plateau, southwest Greenland, Spain, Iceland, and other Northern Hemisphere locations have shown increased climatic variability around the period of the Roman Warm Period (Wang et al., 2012; Bianchi et al., 1999). Temperature records using tree ring chronologies from the Russian Altai and European Alps show the cooling episode defined as the Late Antique Little Ice Age. This period of cooling followed large volcanic eruptions (Buntgen et al., 2016). This period overlaps the Dark Ages Cold period, a period of colder climates spanning the Northern Hemisphere. The contributing factors for this period are under debate, but may involve ice-rafting events, North Atlantic Oscillation, and/or El Nino-Southern Oscillation (Helama et al., 2017). The characteristics of these climate periods depend on the location. For instance, proxy records of the Roman Warm Period indicate that the Mediterranean experienced a wet and humid climate episode (Wang et al., 2012)."

The following has been added to paragraph 2 in section 3.6: "Visual inspection of hemispheric mean temperature data suggests that elevated VA levels from 1080-1240 CE followed elevated Northern Hemisphere temperatures from about 970-1090 CE. Pages 2k Arctic and North American temperature reconstructions show a similar relationship with VA levels (Fig. S11). Elevated VA levels during the Roman Warm Period overlap elevated Arctic temperatures. Elevated VA levels from 1080-1250 CE follow elevated temperatures in the Arctic from about 930-1230 CE and North America from about 750-1150 CE."

Figure S11, showing the Pages 2k Arctic, North American, and European temperature reconstructions, has been added to the supplement.

References added:

Bianchi, G. G. and McCave, 5 I. N.: Holocene periodicity in North Atlantic climate and deep-ocean flow south of Iceland, Nature, 397, 515, doi:10.1038/17362, 1999.

Pages2k: Continental-scale temperature variability during the past two millennia, Nature geoscience, 6, 339, doi:10.1038/NGEO1797, 2013.}

2. "The authors do not compare the Tunu VA record to any other proxy information derived from the Tunu core, other than accumulation, which is surprising as the ice core data would help to better link the fire activity recorded at Tunu to climate variables influencing the site."

[Figure]

Figure S12, showing the Tunu $\delta$18O record, has been added to the supplementary material. An increase in the record during the MCA is referenced in section 3.6. Other Tunu ice core climate proxies are outside the scope of this paper.

3. "Are these blanks the 58 MilliQ water blanks that were previously mentioned? Does the interference also occur in the procedural blanks? (It is unclear if you have both lab and procedural blanks. Please specify exactly how you created your blanks). This peak also occurs in your standards, suggesting that the source of contamination is local, yet you are able to quantify detection limits for pHBA. Please explain."

The blanks referred to are the 58 MilliQ water lab blanks previously mentioned. The source of contamination does appear to be local and likely occurred during sample handling prior to analysis. The source of the contamination has still not been determined. The detection limit for p-HBA was defined using MilliQ water blanks that did not show the contamination.

Section 2.2 has been edited as follows: "This peak was present in most of the ice core samples, several of the 58 MilliQ water blanks, and standards. The detection limit for p-HBA was determined using blanks that did not show this contamination. The presence of this peak in blanks and standards suggests that it is a contaminant that was introduced locally during sample handling. This peak was not observed in previous analyses of p-HBA in other Arctic ice cores (Grieman et al., 2017, 2018)."

4. "Section 2.2: However, more information from the Grieman et al., 2015 method paper is necessary within Section 2.2 of this paper. (Please also change your citations from Grieman et al., 2017, 2018, in review, to Grieman et al., 2015 and 2017 to correctly cite to work to which you refer). For example, what QA/QC measures do you use? Do you apply an internal standard? Readers cannot analyze your interpretations of the data without having a clear indication of the quality of the data."

The method is described in Grieman et al. (2017) and Grieman et al. (2018). Grieman et al. (2015) describes a different method.

The first paragraph of section 2.2 now reads as follows: "Vanillic acid (VA) and para-hydroxybenzoic acid (p-HBA) were analyzed by anion exchange chromatographic separation and tandem mass spectrometric detection with electrospray ionization in negative ion mode (IC-ESI-MS/MS) (Grieman et al., 2017, 2018). The experimental method is described in detail in Grieman et al. (2017). The experimental system is a Dionex AS-AP autosampler, ICS-2100 integrated reagent-free ion chromatograph, and ThermoFinnigan TSQ Quantum triple quadrupole mass spectrometer. VA and p-HBA were detected at mass transitions of m/z 167→108 and m/z 137→93, respectively. Synthetic external standards, ranging in concentration from 0.1-2 ppb, were prepared using reagent grade VA and p-HBA in MilliQ water. These standards were analysed in sequence with ice core samples. The retention times of VA and p-HBA were 11.1 minutes and 11.8 minutes, respectively, with peak width half heights of 0.4 minutes. Detection limits for VA and p-HBA were 0.005 ppb and 0.034 ppb, defined as 3x the standard deviation of MilliQ water blanks (n = 58). 575 ice core samples from the Tunu ice core were analyzed in this study."

5. "Section 3.3: If 17% of the back-trajectories are from North America in the summer, and if 3% are from European ecofloristic zones, and 2% from Siberian ecofloristic zones, where is the source of the rest of the summer back-trajectories?"

The rest of the back trajectories do not transect North American, European, or Siberian ecoflorisitic zones. They primarily originate from the ocean.

The following was added to section 3.3: "The remainder of the trajectories originate over the oceans and did not transect North American, European, or Siberian ecofloristic zones."

6. "Section 3.4 and Figure 8: However, while the relationships between these proxies after 1200 CE certainly are variable, and do not demonstrate any patterns either between themselves or with each other. The only relationship that does seem to exist – and this is really quite stretching the interpretation – is that the correlation coefficients for Tunu VA, NEEM NH4 versus Tunu VA, NEEM BC are out of phase with one another after 1300 CE. It is better to state that after 1300 CE no major relationships exist between these data. Panel 2 is difficult to read. Either separating each proxy or elongating the y-axis will help the reader."

Sentence in section 3.4 changed to: "The results show a strong positive correlation for all of the records from 650-1200 CE but not subsequently."

7. Section 3.5 and Figure S9: You mention "the qualitative similarities between the Tunu VA and western Canadian charcoal record suggest that this may be a source region of the Greenland burning signals, but the correlation is not significant at the 95% confidence interval." However, in examining Figure S9 closely, the only correlation occurs at 1400 CE, while all other peaks and low points are offset from one another. These records do not correlate, and ascribing western Canada as a source region based on this data is overreaching what the data actually demonstrate.

Sentence changed to: "Only the records from western Canada (40°-80°N, 110°-180°W) show any similarity to the Tunu VA record, with a slight increase around 1400."

Removed sentence: "The qualitative similarities between the Tunu VA and western Canadian charcoal record suggest that this may be a source region of the Greenland burning signals, but the correlation is not significant at the 95% confidence interval."

8. Section 3.6 and Figure 9: "The data suggest a positive correlation between North American fire and hemispheric mean temperature." This statement assumes that Tunu VA represents North American fire rather than boreal fire and/or regional fire. The comparison between Tunu VA and temperature for the entire northern hemisphere differs from your plots showing that the back-trajectories are primarily for high latitude regions. Using a temperature record that reflects your source region is a better comparison. (Using a representative temperature record is especially important as the MCA and LIA vary regionally, and therefore influence your time periods of interest). Did you calculate this correlation? Or do the data only "suggest" this correlation? The

peak in Tunu VA and the peak in the NHT anomaly are offset by at least a century. The decreased temperature and lower concentrations of Tunu VA do occur during similar time periods.

The following has been added to paragraph 2 in section 3.6: "Visual inspection of hemispheric mean temperature data suggests that elevated VA levels from 1080-1240 CE followed elevated Northern Hemisphere temperatures from about 970-1090 CE. Pages 2k Arctic and North American temperature reconstructions show a similar relationship with VA levels (Fig. S11). Elevated VA levels during the Roman Warm Period overlap elevated Arctic temperatures. Elevated VA levels from 1080-1250 CE follow elevated temperatures in the Arctic from about 930-1230 CE and North America from about 750-1150 CE."

Figure S11, showing the Pages 2k Arctic, North American, and European temperature reconstructions, has been added to the supplement.

Added Reference:

Pages2k: Continental-scale temperature variability during the past two millennia, Nature geoscience, 6, 339, doi:10.1038/NGEO1797, 2013.

9. Conclusions and abstract: The correlation between Tunu VA, and Neem ammonium and black carbon in the NEEM ice core only exists between 600 to 1200 CE, and so following statement is misleading: "The correlation between Tunu VA, and NEEM ammonium and black carbon in the NEEM ice core is encouraging evidence that a consistent pattern of centennial-scale variability in North American high latitude fire is recorded in Greenland ice, but further measurements on multiple ice cores will be needed to validate this conclusion". A similar sentence is also present in the abstract. This correlation is only true for one part of the record, but absolutely does not apply to the rest of the record.

Sentences in abstract changed as follows: "Analysis using a peak detection method

revealed a positive correlation between vanillic acid in the Tunu ice core and both ammonium and black carbon in the north Greenland NEEM ice core from 600 to 1200 CE. The data provide multiproxy evidence of centennial-scale variability in North American high latitude fire during this time period."

Sentence in conclusion changed as follows: "The correlation between Tunu VA and ammonium and black carbon in the NEEM ice core from 600 to 1200 CE is evidence of centennial-scale variability in North American high latitude fire during this time period. Further measurements on multiple ice cores will be needed to validate this conclusion."

10. Conclusions: The statement "A clear link between the VA variability in Greenland ice and North American sedimentary charcoal is not evident, although a tentative connection to the Quebec region was noted". This statement contradicts your text in Section 3.5 in which you highlight possible similarities (with which the data do not actually demonstrate, as mentioned in the above comment addressing Section 3.5) between the Tunu NA and western Canadian charcoal syntheses. It is essential to omit the conclusion regarding Quebec.

Removed from conclusion: "...although a tentative connection to the Quebec region was noted."

11. Figure 5: The information given by the fraction of ice core samples exceeding the 70th percentile actually detracts from highlighting the difference between the fraction of ice core samples exceeding the set peak threshold in the 65th and 75th percentiles. However, as you use the 70th percentile in section 3.4 and as the basis for the second and third panels of Figure 8, it may be better to only show the 70th percentile. These colors (as in Figure 8) are unfortunately the most difficult to differentiate for people who are color blind, resulting in a figure that is unreadable for many people. Please choose other colors.

Now only use 75th percentile is shown in figures 5 and 8. Text reflects this choice. Color used for ammonium in panel 2 of figure 8 changed. Black line used in panel 3 of

figure 5.

12. Page 3 Lines 32 and 33 (and continuing throughout the paper): PC's and RC's should be changed to PCs and RCs.

Edit complete

13. Page 4, Line 11: Place "of" between "frequency large".

Edit complete

14. Page 4, Line 12: Change "fiers" to "fires".

Edit Complete

15. Page 4, Line 15: Omit the comma after "fluxes".

Edit Complete

16. Page 6, Line 25: "The Canada charcoal record exhibits no linear trend and century scale variability that is not significant at the 95% confidence interval". This is double negative. Do you mean "The Canada charcoal record exhibits neither a linear trend nor century-scale variability that is significant at the 95% confidence interval"?

The sentence has been changed as suggested.

---

## Author Comment (AC2) · 31 Jul 2018

We greatly appreciate the referee's comments. The manuscript has been edited as described below to take them into account.

1. "There should be some discussion of what in particular might be most informative/useful going forward in terms of new proxy comparisons, or modeling, etc."

"Moreover, the influence of different modes of climate variability on the VA record are nearly impossible to disentangle given that the decadal-scale variability in the VA record raises more questions than it addresses – it seems unclear still what exactly is producing the short-term variability in VA (in terms of fire), let alone what would cause the variations in fire themselves IF that is what the VA variations are reflecting."

The following has been added to the conclusion: "At this stage, ice core VA should be regarded as a qualitative tracer because it is not known to what extent the signals reflect paleofire emissions, paleofire frequency, or changes in air mass transport and deposition. Further work comparing VA in shallow ice cores to satellite measurements and modeling of fires during recent decades would improve our understanding of the origins of the VA signals in Greenland ice."

2. "Furthermore, how might we better understand the source areas or relationships between VA and other fire proxies from ice (e.g. what benefits does VA provide in comparison with BC, ammonium, levoglucosan, etc.)? Where are the greatest sources of uncertainty? Clarifying the limitations of the existing study and adding some thoughts about 'next steps' would be helpful for those outside the ice core community in particular."

The following description of other ice core biomass burning proxies has been added to the introduction: "A range of ice core proxies have been used to reconstruct biomass burning (Legrand et al., 2016; Rubino et al., 2015). Stable isotope ratios of ice core methane ($\delta$13CH4) have been used to infer global biomass burning, using end member isotopic compositions of methane sources (Ferretti et al., 2005; Mischler et al., 2009). Elevated ammonium concurrent with elevated levels of other chemicals has been used to reconstruct regional biomass burning. The difficulty of using ice core ammonium is that it is derived from several sources (Rubino et al., 2015). Ice core black carbon has been used as a tracer for preindustrial burning. Differences between these records could be due to variability in combustion conditions and transport (Rubino et al., 2015). The incomplete combustion of biomass produces organic aerosols. A large percentage of the biomass burning-derived organic aerosol is composed of levoglucosan, which is emitted by all plant matter containing cellulose (Simoneit et al., 1999). However, levoglucosan has the potential for rapid degradation during atmospheric transport (Hennigan et al., 2010; Hoffmann et al., 2010; Slade et al., 2013)."

Added References:

Ferretti, D. F., Miller, J. B., White, J. W. C., Etheridge, D. M., Lassey, K. R., Lowe, D. C., Meure, C. M. M., Dreier, M. F., Trudinger, C. M., van Ommen, T. D., and Langenfelds, R. L.: Unexpected Changes to the Global Methane Budget over the Past 2000 Years, Science, 309, 1714–1717, doi:10.1126/science.1115193, 2005.

Hennigan, C. J., Sullivan, A. P., Collett, J. L., and Robinson, A. L.: Levoglucosan stability in biomass burning particles exposed to hydroxyl radicals, Geophysical Research Letters, 37, L09 806, doi:10.1029/2010GL043088, 2010.

Hoffmann, D., Tilgner, A., Iinuma, Y., and Herrmann, H.: Atmospheric stability of levoglucosan: a detailed laboratory and modeling study., Environmental Science and Technology, 44, 694–699, doi:10.1021/es902476f, 2010.

Sapart, C. J., Monteil, G., Prokopiou, M., van deWal, R. S.W., Kaplan, J. O., Sperlich, P., Krumhardt, K. M., van der Veen, C., Houweling, S., Krol, M. C., Blunier, T., Sowers, T., Martinerie, P., Witrant, E., Dahl-Jensen, D., and Rockmann, T.: Natural and anthropogenic variations in methane sources during the past two millennia, Nature, 490, 85–88, 10.1038/nature11461, 2012.

Slade, J. H. and Knopf, D. A.: Heterogeneous OH oxidation of biomass burning organic aerosol surrogate compounds: assessment of volatilization products and the role of OH concentration on the reactive uptake kinetics., Physical Chemistry Chemical Physics, 15, 5898–915, doi:10.1039/C3CP44695F, 2013.

3. "Section 3.3 – This section is describing methods but is located in the Results section – it seems more appropriate to methods. In addition, the 17% of the summer back trajectories reaching North America is quite low – where do all the trajectories that are not accounted for come from?"

The first paragraph of section 3.3 has been moved to the methods section (Section 2.3). They primarily originate from the ocean. A low percentage of the trajectories also transect Europe and Siberia.

4. "Section 3.6 also needs work. The methods referenced are entirely absent from the "Methods" section again and need to be moved and expanded there to understand what exactly was done (Pg. 7 line 19 is insufficient)."

This section describes a comparison to several climate proxy records presented in the publications referenced. The methods used to produce these records are described in the referenced publications and we think it is beyond the scope of this paper to describe them here.

5. "Furthermore, the features that are most robust in the VA record – high values during the RWP and MCA, and low values during the LALIA and LIA – are still not well understood. The RWP itself is not a well-known or widespread climate feature, so more information and background about that could be provided instead of discussion of possible climate modes that might control fire."

The following has been added to the first paragraph of section 3.6 to define the lesser-known climate periods: "Climate proxies from the North Sea, the Qinghai–Tibet Plateau, southwest Greenland, Spain, Iceland, and other Northern Hemisphere locations have shown increased climatic variability around the period of the Roman Warm Period (Wang et al., 2012; Bianchi et al., 1999). Temperature records using tree ring chronologies from the Russian Altai and European Alps show the cooling episode defined as the Late Antique Little Ice Age. This period of cooling followed large volcanic eruptions (Buntgen et al., 2016). This period overlaps the Dark Ages Cold period, a period of colder climates spanning the Northern Hemisphere. The contributing factors for this period are under debate, but may involve ice-rafting events, North Atlantic Oscillation, and/or El Nino-Southern Oscillation (Helama et al., 2017). The characteristics of these climate periods depend on the location. For instance, proxy records of the Roman Warm Period indicate that the Mediterranean experienced a wet and humid climate episode (Wang et al., 2012)."

Reference added:

Bianchi, G. G. and McCave, 5 l. N.: Holocene periodicity in North Atlantic climate and deep-ocean flow south of Iceland, Nature, 397, 515, doi:10.1038/17362, 1999.

6. "I would rather see more investigation into understanding the potential relationships between the VA record and other fire proxies than a detailed investigation into the interannual climate controls on a very uncertain biomass burning reconstruction."

The following has been added to section 3.4: "Levoglucosan and black carbon in the NEEM ice core from Northern Greenland are elevated from 200-600 CE and 100-700 CE, respectively (Fig. S8) (Zennaro et al., 2014). These periods overlap the period of elevated Tunu VA from 280-400 CE. They are still elevated 200-300 years after the Tunu VA record has declined. There is also an overlapping peak in the GISP2 ice core ammonium record from 320-330 CE (Chylek et al., 1995). NEEM levoglucosan, black carbon, and ammonium are also elevated from 1000-1200 CE, 1000-1600 CE, and 1200-1500 CE respectively, at about the same time as the peak in the Tunu VA record from 1080-1240 CE (Legrand et al., 2016; Zennaro et al., 2014). This peak is slightly earlier than a period of elevated ammonium, oxalate, and potassium in an ice core from the Eclipse ice field in western Canada from 1240-1410 CE (Yalcin et al., 2006). Periods of elevated burning in Greenland and Canadian ice core records after 1240 CE are not pronounced in the Tunu VA record. These periods include elevated NEEM levoglucosan from 1500-1700 CE, 2D and GISP2 ammonium from 1790-1810 CE and 1830-1910 CE, and periods of elevated burning in the Mt. Logan and Eclipse ice cores from western Canada in the 18th-20th centuries (Whitlow et al., 1994; Yalcin et al., 2006; Zennaro et al. 2014)."

Figure S8 showing a timeline of elevated periods of biomass burning in ice core records was also added to the supplement.

Added References:

Chylek, P., Johnson, B., Damiano, P. A., Taylor, K. C., and Clement, P.: Biomass burning record and black carbon in the GISP2 Ice Core, Geophysical Research Letters, 22,

89–92, doi:10.1029/94GL02841, 1995.

Whitlow, S., Mayewski, P., Dibb, J., Holdsworth, G., and Twickler, M.: An ice-core-based record of biomass burning in the Arctic and Subarctic, 1750–1980, Tellus B: Chemical and Physical Meteorology, 46, 234–342, doi:10.3402/tellusb.v46i3.15794, 1994.

Yalcin, K., Wake, C. P., Kreutz, K. J., and Whitlow, S. I.: A 1000-yr record of forest fire activity from Eclipse Icefield, Yukon, Canada, The Holocene, 16, 200–209, doi:10.1191/0959683606hl920rp, 2006.

7. "Pg 2. VA is described as resulting from the combustion of lignin. McConnell et al. (2007) indicated that conifers in particular are expected to be the primary source of VA – it seems worth noting this again, and also providing more information that might be useful based on the extensive previous research on VA that is cited. E.g. what is the expected atmospheric residence time of VA?"

The following has been added to paragraph 4 of the introduction: "These studies have shown that North American and European conifer and deciduous tree species produce vanillic acid (VA) (Simoneit, 2002, Oros and Simoneit et al. 2001a, b, Iinuma et al., 2007). VA is observed in atmospheric aerosols in the Arctic and Antarctic after long-distance transport (Zangrando et al., 2013, 2016}. The lifetime of aromatic acids in the gas phase is on the order of a day due to oxidation by the hydroxyl radical. However, modeling studies suggest that in aerosols these compounds may be shielded from oxidation, resulting in atmospheric lifetimes of several days (Donahue et al., 2013). Laboratory and field studies have also shown decreased volatility of low molecular weight organic acids in aerosol form as a result of interactions with sea salt and other cations. Such studies have not yet been carried out on aromatic acids (Häkkinen et al., 2014, Laskin et al., 2012)."

Added References:

Donahue, N., Chuang,W., Epstein, S., Kroll, J.,Worsnop, D., Robinson, A., Adams,

P., and Pandis, S.:Why do organic aerosols exist? Understanding aerosol lifetimes using the two-dimensional volatility basis set, Environmental Chemistry, 10, 151–157, doi:10.1071/EN13022, 2013.

Häkkinen, S. A. K., McNeill, V. F., and Riipinen, I.: Effect of Inorganic Salts on the Volatility of Organic Acids, Environmental Science & Technology, 48, 13 718–13 726, doi:10.1021/es5033103, 2014.

Laskin, A., Moffet, R. C., Gilles, M. K., Fast, J. D., Zaveri, R. A., Wang, B., Nigge, P., and Shutthanandan, J.: Tropospheric chemistry of internally mixed sea salt and organic particles: Surprising reactivity of NaCl with weak organic acids, Journal of Geophysical Research: Atmospheres, 117, D15 302, doi:10.1029/2012JD017743, 2012.

Zangrando, R., Barbaro, E., Zennaro, P., Rossi, S., Kehrwald, N. M., Gabrieli, J., Barbante, C., and Gambaro, A.: Molecular markers of biomass burning in Arctic aerosols, Environmental science & technology, 47, 8565–8574, doi:10.1021/es400125r, 2013.

Zangrando, R., Barbaro, E., Vecchiato, M., Kehrwald, N. M., Barbante, C., and Gambaro, A.: Levoglucosan and phenols in Antarctic marine, coastal and plateau aerosols, Science of The Total Environment, 544, 606 – 616, doi:10.1016/j.scitotenv.2015.11.166, 2016.

8. "Pg 4 Line 6: Were higher thresholds tried? Would thresholds in the 80s or 90s produce very different results? Why did you choose this range?"

The following has been added to paragraph 3 of section 3.4: "Increasing the threshold from the 80th-95th percentiles reduces the number of peaks, but does not significantly alter their timing."

9. "Pg 4 Line 8: remove "Clearly" and explain what 'these' refers to."

Edit complete and removed sentence: "These are robust features of the Tunu VA record."

10. "Pg 4 Line 12: should say 'fires'"

Edit Complete

11. "Pg 4 line 20: define the terms in the equation and make your words consistent with the equation. Pg 4 Lines 26-80 – the equation and its description are not rendering properly (esp. check the capitalization and subscripting; vd, rscav)."

Equation and equation term definitions edited as follows:

"Ftotal = Fdry + Fwet = Cair Vd + Cair Rscav PH2O,

where Ftotal is the total VA flux, Fdry is the VA dry deposition flux, Fwet is the VA wet deposition flux, where Cair is the concentration of VA in the atmosphere, Vd is a dry deposition velocity for the VA-containing aerosols, Rscav is the wet deposition scavenging ratio, and PH2O is the snow precipitation rate (Saltzman et al., 1997)."

12. "Nor is there discussion of the more recent variations and how they compare with historical data, which is a large omission that needs to be addressed, especially given the 2.5-year temporal resolution of the ice core record noted in Section 2.1."

"Fig. 4 – Please discuss this figure and how it relates to the known fire history of NA. Fire history data are available from tree-ring (e.g. Giardin's papers, the Canadian Large Fire Database) and historical studies and the data raise important concerns that warrant more discussion. Specifically, the high VA values in the 1960's and 70's are puzzling given that this pattern is directly opposite what historical records show (e.g. Mouillot and Field, 2005; Mouillot et al. 2006), where burning was very low in the middle part of the century. Many more recent NA fire records generally show that burning has increased most rapidly from low levels since the 1980's (NIFC, Littell et al. 2009). Is it possible that VA could at times have been influenced by some intensive industrial or lumber/logging-related processes that might have been occurring during this time in NA? In any case, in light of these data it is good that the authors state that VA is best considered a qualitative proxy at this stage and requires further exploration – this

acknowledgement is appreciated."

The following has been added to section 3.1: "Burned area in the Boreal forest of North America was high in the early 1900s, declined into the middle of the twentieth century, and began to increase again beginning in the 1960s (Mouillot and Field, 2005). The short-term period of elevated Tunu VA from 1955-1985 does not mirror the trend in this burned area record. A record of large boreal wildfires in Canada, with burned areas exceeding 200 ha, shows periods of large wildfires around 1960, the late 1970s to the early 1980s, around 1990, and the mid-1990s (Stocks et al., 2003). The similarity between the Tunu VA record and large boreal fire record from Canada suggests that that twentieth century VA record may be showing large fires. These large fires represent 3.1% of the number of Canadian fires from 1959-1997, but 97% of the area burned (Stocks et al., 2003)."

Added References:

Mouillot, F. and Field, C. B.: Fire history and the global carbon budget: a 1_ 1 fire history reconstruction for the 20th century, Global Change Biology, 11, 398–420, doi:10.1111/j.1365-2486.2005.00920.x, 2005.

Stocks, B., Mason, J., Todd, J., Bosch, E., Wotton, B., Amiro, B., Flannigan, M., Hirsch, K., Logan, K., Martell, D., et al.: Large forest fires in Canada, 1959–1997, Journal of Geophysical Research: Atmospheres, 107, FFR–5, doi:10.1029/2001JD000484, 2002.